# HortNet417v1—A Deep-Learning Architecture for the Automatic Detection of Pot-Cultivated Peach Plant Water Stress

**DOI:** 10.3390/s21237924

**Published:** 2021-11-27

**Authors:** Md Parvez Islam, Takayoshi Yamane

**Affiliations:** 1Research Center for Agricultural Robotics, NARO, Tsukuba 3050856, Japan; p.islam@affrc.go.jp; 2Research Center for Agricultural Information Technology and National Institute of Fruit Tree and Tea Science, NARO, Tsukuba 3050856, Japan

**Keywords:** deep learning, network architecture, stem water potential, plant water stress, classification

## Abstract

The biggest challenge in the classification of plant water stress conditions is the similar appearance of different stress conditions. We introduce HortNet417v1 with 417 layers for rapid recognition, classification, and visualization of plant stress conditions, such as no stress, low stress, middle stress, high stress, and very high stress, in real time with higher accuracy and a lower computing condition. We evaluated the classification performance by training more than 50,632 augmented images and found that HortNet417v1 has 90.77% training, 90.52% cross validation, and 93.00% test accuracy without any overfitting issue, while other networks like Xception, ShuffleNet, and MobileNetv2 have an overfitting issue, although they achieved 100% training accuracy. This research will motivate and encourage the further use of deep learning techniques to automatically detect and classify plant stress conditions and provide farmers with the necessary information to manage irrigation practices in a timely manner.

## 1. Introduction

Water management is significant for controlling fruit tree growth, yield, and fruit quality [1]. Irrigation methods for fruit trees are often determined by soil drying; however, the condition of part of the soil does not always match that of the whole soil in the root zone, because fruit tree roots are distributed at various depths depending on the soil type and floor management [2]. Plant-based irrigation is a significantly better method than soil-based estimation to save water, maintain optimal plant growth, and improve fruit quality [3]. Stem water potential is a widely accepted method to measure plant water status [4,5]. However, these measurements are destructive and labor intensive. Therefore, a low-cost and non-destructive method is needed.

Due to the tremendous development and reduction in cost of image acquisition, data storage, and computing systems, the computer vision technique has become a popular tool for deep learning (DL). DL in general is the successor of machine learning with many hidden layers and massive training data [6] with the capability of an automatic acquiring for hierarchical feature and learning via non-linear filters from the bottom and thereafter producing decision-making or classification at the top [7]. Deep learning emphasizes the depth of the deep network structure to learn complex features with a higher level of accuracy, enabling a network to solve problem-specific tasks such as object detection, semantic segmentation, and image analysis. It has been reported that many well-known DL architectures, such as LeNet, AlexNet, VGG, ResNet, GoogleNet, MobileNet, Inception, and SqueezeNet, are widely used to identify and classify leaf diseases [8]. In agriculture, it is difficult to separate the object from the background because of their similarity. On the other hand, the dynamic nature of the background in orchards and farmland creates challenges for the classification task.

Wakamori et al. [9] proposed a multimodal neural network with long short-term memory layers. Their network requires physiological and meteorological data to estimate plant water stress with a 21% mean absolute and root-mean-squared error. Their target plant was a rockwool-cultivated tomato grown in a greenhouse with a very short and shallow rootzone. For physiological data, they used images of leaf wilting. The leaf-wilting phenomenon is mostly related to daytime transpiration [10]. This might be helpful in plants that grow inside a greenhouse. However, leaf wilt also depends on leaf age and weather conditions, such as rainfall and wind. Our target plant is the peach tree, which is long and deep rooted and grown mostly in open-field conditions. Soil-moisture conditions mostly depend on the soil type, and both moisture and soil type are non-uniform in depth and distance from the plant. The water uptake capacity also varies from plant to plant, and transpiration varies among different parts of the peach tree.

Kamarudin et al. [11] conducted a comprehensive review of various plant water-stress assessment methods and reported that most of the existing deep-learning solutions are based on soil moisture estimation, soil water modelling, evapotranspiration estimation, evapotranspiration forecasting, plant water-stress estimation, and plant water-stress identification. All of these methods use sensory measurement data for machine learning analysis, which mostly depends on the quality of the data and the specific target application [12]. However, deep learning is mostly used in disease detection and yield prediction.

The author proposed a new DL architecture with several components (convolutions, batch normalization, ReLU, max pooling, depth concatenation, element wise addition, dropout, fully connected layers, softmax, classification output layer, etc.). In some literature [13,14], it is explained that deepening the DL network could increase classification accuracy with a noisy background. Therefore, the depth of the DL architecture is fixed to 417 layers and accurately optimized based on training performance. The aim of this work is to demonstrate the HortNet417v1 (horticulture network 417 version 1) performance for the automatic detection of peach plant leaf water stress under various environmental conditions.

## 2. Materials and Methods

Nine 9-year-old peach trees (*Prunus persica* (L.) Batsch cv. ‘Akatsuki’ grafted onto ‘Ohatsumomo’ peach), were planted in 60 L plastic pots in a greenhouse at the National Institute of Fruit Tree and Tea Science, National Agriculture and Food Research Organization (NARO) in Tsukuba, Ibaraki, Japan. This experiment was conducted from 27 May to 19 June 2020. Full bloom was 28 March and the growth stage of these trees was from 65 (stage II of the fruit-developing stage) to 88 days (stage III of that two weeks before harvest) after full bloom. The cultivation process is shown in Figure 1.

The pot cultivation method was used to simulate the water stress condition. The stem water potential method verified the stress conditions of each peach tree. Three irrigation methods were applied to create different stress conditions: (1) Wet treatment: when the soil water potential reached −0.005 MPa, three potted trees were automatically irrigated to maintain moist conditions. (2) Dry treatment: when the soil water potential reached −0.050 MPa, three potted trees were automatically irrigated to maintain that level. But during the experiment, the irrigation was stopped four times until the soil water potential reached about −0.070 MPa.

A tensiometer (DM-8HG, Takemura Denki Seisakusho Co., Ltd., Tokyo, Japan) was used to monitor the soil water potential at a depth of 15 cm every 10 min and photographs were taken of these tress under various stress conditions and different water conditions. Two matured leaves per tree were enclosed in a small hermetic aluminum foil bag at least 30 min before the measurement time. These leaves reached equilibrium with the main stem of the tree; therefore, their water potential represented the whole tree. This method is known to be stable to estimate the water condition of the whole tree [15,16]. The stem water potential of each tree was measured 22 times under various stress conditions from 27 May to 19 June 2020. A pressure chamber (Model 600, PMS Instrument Company, Albany, OR, USA) was used for this experiment. Photographs were taken using a smart phone equipped with a camera (iPhone 11, Apple, Cupertino, CA, USA) (Figure 2).

We randomly take around 100 image per tree to ensure the data diversity under various illumination conditions (shaded/sunlit), air temperatures, proximate distances, and points of view (direction and height). A total of 25,000 images were obtained during the experiment. Water-stress conditions were divided into five classes for subsequent analysis by stem water potential: (1) no stress, over −0.5 MPa; (2) low stress, −0.5 to −1.0 MPa; (3) moderate stress, −1.0 to −1.5 MPa; (4) high stress, −1.5 to −2.0 MPa; (5) very high stress, under −2.0 MPa.

The proposed network architecture of HortNet417v1 represents a semantic segmentation-based network (convolution layers) and a classification-based network (softmax), as shown in Figure 3.

The higher-level image features extracted from input image have multiple and smaller filter sizes of 3 × 3 × 3 × 32 pixels in the second layer of the network. The convolution in the second layer has a strong generalization ability that improves the network initial learning performance even when the input images are conglutinated with various objects [17].

The ReLU (rectified linear unit) activation function adds non-linearities to the proposed network, and converts the value of each input element that is less than zero to zero by performing a threshold operation [18]. The ReLU layer also converts the total weighted input from the node to the output, which overcomes the vanishing gradient problem. This also enables the model to learn faster and perform with higher accuracy. However, ReLU is not continuously differentiable and sometimes leads to cause the dying ReLU problem (neuron death). To prevent this, a LeakyReLU is added, where any input value less than zero is multiplied by a fixed scalar (fixed manually). This will not saturate with the positive value of the input, so it can prevent the gradient from exploding/disappear during the gradient descent process.

A ClippedReLU reduces the feature value of the output from becoming too large [19]. The batch-normalization layer transforms each input in the mini-batch, subtracts the batch mean, and then divides the mini-batch by the standard deviation [20]. This layer normalized the output from a previous activation layer and increased the network stability.

In addition, in order to avoid the use of mini-batch dimensions, we introduce a group normalization layer, which divides the input channels into groups and normalizes across each group [21]. The max pooling layer simplifies the complexity of the network by compressing features for fast computing, and extracting main features, thereby ensuring the invariance of feature positions and rotations [17,22]. The average pooling layer creates a downsampled (pooled) feature by dividing the input into rectangular regions, computing the average values of each region, adding a small amount of translation invariance, and extracting smooth features, whereas max pooling mostly extracts features like edges [23,24].

The dropout in the 404th layer randomly drops 10% of neurons in order to prevent the overfitting issue. The last convolution of the 411th layer has 64 filters, with batch normalization and ReLU activation, and then global average pooling. We added global average pooling to the 414th layer (1 × 1 × 64) before the fully connected layer. This layer downsamples the height and width dimensions of the input, and reduces the total number of parameters without sacrificing performance and minimizes overfitting. After multiple rounds of convolution and pooling, all abstract features are integrated into a vector through a fully connected layer. This layer has five outputs, corresponding to five classes that feed into the softmax layer for calculating the probability of the output classification layer. These expanded features passed to the classification layer for classification [25]. Several performance metrics, such as training/validation/test accuracy, confusion matrices, and validation loss and training time, evaluate the proposed deep-learning network architecture. The confusion matrices are given in terms of percentage and absolute number. Therefore, the depth of the DL architecture is fixed to 417 layers and accurately optimized based on training performance. The statistics of the HortNet417v1 architecture are shown in Table 1.

The learning parameters applied to train HortNet417v1 were validation frequency: 50; validation patience: inf; mini-batch size: 260; maximum epoch: 50; learn rate schedule: piecewise; shuffle: every epoch; initial learn rate: 0.001; learn rate drop period: 10; learn rate drop factor: 0.1; L2 regularization: 0.0001; sequence length: longest; sequence padding value: 0; sequence padding direction: right; and epsilon: 1.00E-08. The adaptive moment estimation (ADAM) was used to optimize the network weights. All analyses were run using the Supercomputer FUJITSU SHIHO equipped with TESLA V100 -SXM2 32GB and CUDA version 10.2, Deep Learning and Parallel computing Toolbox (Matlab R2020a).

For comparative analysis, we used modified pretrained network NasNet-Mobile, ResNet-50, Xception, ShuffleNet, SqueezeNet, GoogleNet, and MobileNetv2, as shown in Table 2.

We designed HortNet417v1 in such a way that allows it to be used in both low-powered mobile and high-powered fixed devices. Several performance metrics, such as training accuracy (TA), validation accuracy (VA), test accuracy (TeA), confusion matrix, training loss (TL), validation loss (VL), sensitivity (Equation (1)), specificity (Equation (2)), precision (Equation (3)), F1 score (Equation (4), and Matthews correlation coefficient (MCC, Equation (5)) are used to evaluate HortNet417v1 network efficiency. However, $\Delta Accuracy_{Tr-Val}$ in Equation (6) and $\Delta Loss_{Tr-Val}$ in Equation (7) are used to evaluate HortNet417v1 network overfitting issue. The same performance metrics are also evaluated on NasNet-Mobile, ResNet-50, Xception, ShuffleNet, SqueezeNet, GoogleNet, and MobileNetv2 for comparative analysis. Visualization of the predicted stress condition for evaluating the accuracy based on validation (6266 images) and test (500 images) data are also provided in the Results Section 3 and Discussion Section 4.
(1)$$\mathrm{Sensitivity}=\frac{\mathrm{TP}}{\mathrm{TP}+\mathrm{FN}}$$
where, TP—true positive, FN—false negative, worst value = 0, best value = +1.
(2)$$\mathrm{Specificity}=\frac{\mathrm{TN}}{\mathrm{TN}+\mathrm{FP}}$$
where, TN—true negative, worst value = 0, best value = +1.
(3)$$\mathrm{Precision}=\frac{\mathrm{TP}}{\mathrm{TP}+\mathrm{FP}}$$
where, worst value = 0, best value = +1.
(4)$$F1-\mathrm{score}=2\times \frac{\;\left(\mathrm{Sensitivity}\;\times \mathrm{Precision}\right)\;}{\left(\mathrm{Sensitivity}\;+\mathrm{Precision}\right)}$$
where, worst value = 0, best value = +1.
(5)$$\mathrm{MCC}=\frac{\mathrm{Cov}\left(c,l\right)}{\sigma_{c}\times \sigma_{l}}=\frac{\mathrm{TP}\times \mathrm{TN}-\mathrm{FP}\times \mathrm{FN}}{\sqrt{\left(\mathrm{TP}+\mathrm{FP}\right)\times \left(\mathrm{TP}+\mathrm{FN}\right)\times \left(\mathrm{TN}+\mathrm{FP}\right)\times \left(\mathrm{TN}+\mathrm{FN}\right)}}$$
where, $\mathrm{Cov}\left(c,l\right)$ is the covariance of the true classes *c* and predicted labels *l*, $\sigma_{c}$ and $\sigma_{l}$ are the standard deviations, worst value = −1, best value = +1.
(6)$$\Delta {\mathrm{Accuracy}}_{\mathrm{Tr}-\mathrm{Val}}=\frac{\mathrm{TA}\;–\;\mathrm{VA}}{100}\;\times 100$$
(7)$$\Delta {\mathrm{Loss}}_{\mathrm{Tr}-\mathrm{Val}}=\frac{\mathrm{TL}\;–\;\mathrm{VL}}{100}\times 100$$

We employed a network visualization feature using the deep dream function and compute the network layer activation to visualize the activation of different layers of the network. This also demonstrates how networks identify and learn features in different depths of the network. The t-SNE (t-distributed stochastic neighbor embedding) algorithm is used to reduce the high dimensionality and visualize the low-dimension data in a way that respects the similarities between data points [33]. We use occlusion sensitivity and locally interpretable model-agnostic explanation (LIME) techniques to predict the result evaluation (what/why/how/which/where). The occlusion sensitivity technique demonstrates what input image features are used by the network to make a classification decision and helps to identify the reason behind the network misclassification decision [34]. The LIME technique generates synthetic data from the input, classifying the synthetic data using the net and then using a simpler, more interpretable machine-learning classification/regression model to fit the classification results [35].

Figure 4 shows the pattern of image datasets for the training, validation, and testing of HortNet417v1. In this experiment, 70% of the randomly selected images from the image dataset were used for training and 30% for validation purposes. An independent 500-image dataset (100 images/stress condition) was used for testing purposes. The image dataset was augmented, including random X and Y reflection, to prevent overfitting and to generalize the model for better network learning. A total of 43,866 augmented images were used to train the HortNet417v1 network.

Figure 5 visualizes the experimental step (from dataset preparation to networks training, model evaluation and prediction) for network analysis.

## 3. Results

### 3.1. Performance Evaluation

Table 3 shows the result of several performance metrics for evaluating HortNet417v1 network efficiency. This comparative analysis shows that HortNet417v1 can classify stress conditions with the test accuracy as high as NasNet-Mobile/Resnet-50/Xception/ShuffleNet/MobileNetv2 and higher than GoogleNet or SqueezeNet. From Table 3, we can see that Xception, ShuffleNet, and MobileNetv2 reached the convergence stage after 29.35, 29.23, and 19.3 min of network training, then stopped at 25, 24, and 26 max. epochs, respectively, with 100% training accuracy for all three networks, lower validation accuracy of 96.18%, 92.60% and 94.13% and slightly higher test accuracy of 97.20%, 93.60% and 95.40%. This trend demonstrated that models with higher training accuracy can be overfit by memorizing properties of the training set but failed to predict with validation or test data with same accuracy level [36]. The class imbalance problem of the training dataset might also affect the lower training loss (2%, 3% and 2%) and higher validation loss (11%, 21%, 20%) for Xception, ShuffleNet, and MobileNetv2, respectively. However, this class imbalance problem did not affect our proposed model HortNet417v1 network performance which shows very close training and validation loss of 21% and 20%, respectively.

NasNet-Mobile and ResNet-50 reached their convergence stage after 225.58 and 26.35 min at 25 max epochs with 98.50% and 98.85% training accuracy, respectively. SqueezeNet and GoogleNet achieved their convergence stage after 6.54 and 1.31 min of network training, then stopped at 27 and 3 max epochs with only 58.85% and 28.85% training accuracy, respectively. Our proposed HortNet417v1 network reached its convergence stage after 213 min of training, then stopped at 36 max epochs with 90.77% training accuracy.

From Figure 6, it is clear that the highest $\Delta Accuracy_{Tr-Val}$ is observed with ShuffleNet (7.4%), MobileNetv2 (5.87%), ResNet-50 (4.29%), NasNet-Mobile (2.4%), and Xception (3.82%), while a negative $\Delta Accuracy_{Tr-Val}$ is found with GoggleNet (−1.23%) and SqueezeNet (−0.31%). The lowest $\Delta Accuracy_{Tr-Val}$ is observed with the HortNet417v1 (0.25%) network, and this demonstrated that the network learns without any overfitting issue. On the other hand, the highest $\Delta Loss_{Tr-Val}$ is observed with GoogleNet (0.03%) and SqueezeNet (0.02%), and the negative $\Delta Loss_{Tr-Val}$ is observed with MobileNetv2 (−0.18%), ShuffleNet (−0.18%), ResNet-50 (−0.12%), Xception (−0.09%), and NasNet-Mobile (−0.08%). The lowest $\Delta Loss_{Tr-Val}$ is also observed with the HortNet417v1 (0.01%) network. In both cases (Δ differences between training and validation accuracy and Δ differences between training and validation loss), the network demonstrates stability, and it is possible to improve network performance by adding more data.

It can be seen from the confusion chart in Figure 7a, with a validation dataset, that the higher classification accuracy of 96.5% was achieved for the very high stress condition, followed by no stress (92.6%), moderate stress (87.1%), low stress (89.7%) and high stress (93.2%).

However, the confusion chart in Figure 7b, with a test dataset, also demonstrated that this accuracy increased from 92.6% to 98.9% (very high stress), from 92.6% to 96.9% (no stress), and from 87.1% to 90.3% (moderate stress); meanwhile, it fell slightly from 89.7% to 89.0% (low stress) and from 93.2% to 90.5% (high stress). This performance can be improved by adding more data patterns with a training dataset.

Figure 8 shows the performance indicators of each single class. Even when the data of each class is unbalanced, the HortNet417v1 network achieves the best value for all individual classes.

### 3.2. Visualization of the Predicted Stress Condition for Evaluating Accuracy Based on Test Data

Figure 9a shows 20 randomly selected predicted images to evaluate the accuracy based on 6266 validation and 500 image data sets. The network is perfectly classified based on the validation and test images, and in both cases, as shown in Figure 9a,b, only one misclassification occurred. This issue was caused by the similar plant responses under low and moderate stress conditions (low stress: −0.5 to −1.0 MPa; moderate stress: −1.0 to −1.5 MPa). By adding various patterns with the training dataset for low and moderate stress conditions, especially in a range from −0.8 MPa to −1.2 MPa, it is possible to overcome this problem.

### 3.3. Visualization of the Network Feature and Layer Activations

Figure 10 Visualization of the network feature using deep dream (i, iii, v, vii) and layer activations (ii, iv, vi, viii) at different depths of the HortNet417v1 network.

#### 3.3.1. Convolution Layer

In the shallow depth of the network (layer 2), the input image size is 240 × 240 with 32 filters (Figure 10a(i)), and the weight and bias for learnable parameters are 3 × 3 × 3 × 32 and 1 × 1 × 32, respectively. In the middle depth (139th layer), the input image size is 15 × 15 with 256 filters (Figure 10a(iii)), and the weight and bias for learnable parameters are 3 × 3 × 128 × 256 and 1 × 1 × 256, respectively. In the deep depth (layer 313), the image size is 8 × 8 with 512 filters (Figure 10a(v)), and the weight and bias for learnable parameters are 3 × 3 × 512 × 512 and 1 × 1 × 512, respectively. In the final convolution layer (411th), the image size is 1 × 2 with 64 filters (Figure 10a(vii)), and the weight and bias for learnable parameters are 3 × 3 × 128 × 64 and 1 × 1 × 64, respectively. The learnable feature is the response of the CNN (convolution neural network) layer to an input, as shown in Figure 10a(ii,iv,vi,viii).

#### 3.3.2. Batch-Normalization Layer

In the shallow depth layer (3rd), the input image size is 240 × 240 with 32 filters (Figure 10b(i)), and the offset and scale for learnable parameters are 1 × 1 × 32 and 1 × 1 × 32, respectively. In the middle depth layer (142th), the input image size is 15 × 15 with 256 filters (Figure 10b(iii)), and the offset and scale for learnable parameters are 1 × 1 × 256 and 1 × 1 × 256, respectively. In the deep depth (layer 316), the image size is 8 × 8 with 512 filters, and the offset and scale for learnable parameters are 1 × 1 × 512 and 1 × 1 × 512 (Figure 10b(v)), respectively. In the final convolution (layer 412), the image size is 1 × 2 with 64 filters (Figure 10b(vii)), and the offset and scale for learnable parameters are 1 × 1 × 64 and 1 × 1 × 64, respectively. The learnable feature is the response of the batch-normalization layer, as shown in Figure 10b(ii,iv,vi,viii).

#### 3.3.3. Rectified Linear Unit (ReLU) Layer

In the shallow depth layer (4th), the input image size is 240 × 240 with 32 filters (Figure 10c(i)). In the middle depth (148th layer), the input image size is 8 × 8 with 512 filters (Figure 10c(iii)). In the deep depth layer (324th), the image size is 8 × 8 with 512 filters (Figure 10c(v)). In the final ReLU (layer 413), the image size is 1 × 2 with 64 filters (Figure 10c(vii)). The learnable feature is the response of the ReLU layer, as shown in Figure 10c(ii,iv,vi,viii).

#### 3.3.4. LeakyReLU

In the shallow depth (19th layer), the input image size is 240 × 240 with 32 filters (Figure 10d(i)). In the middle depth layer (221th layer), the input image size is 8 × 8 with 512 filters (Figure 10d(iii)). In the deep depth layer (327th), the image size is 8 × 8 with 512 filters (Figure 10d(v)). In the final LeakyReLU (layer 407th), the image size is 1 × 2 with 256 filters (Figure 10c(vii)). The learnable feature is the response of the LeakyReLU layer to an input, as shown in Figure 10d(ii,iv,vi,viii).

#### 3.3.5. ClippedReLU

In the shallow depth (layer 26th), the input image size is 120 × 120 with 64 filters (Figure 10e(i)). In the middle depth layer (244th), the input image size is 8 × 8 with 512 filters (Figure 10d(iii)). In the final ClippedReLU (layer 339th), the image size is 8 × 8 with 512 filters (Figure 10d(v)). Figure 10e(ii,iv,vi) shows the learnable feature which is the response of a ClippedReLU layer to an input.

#### 3.3.6. Average Pooling

In the shallow depth (layer 15th), the input image size is 240 × 240 with 32 filters (Figure 10f(i)). In the middle depth (120th layer), the input image size is 60 × 60 with 256 filters (Figure 10f(iii)). In the deep depth layer (160th), the image size is 8 × 8 with 512 filters (Figure 10f(v)). The learnable feature is the response of the average pooling layer to an input, as shown in Figure 10f(ii,iv,vi).

#### 3.3.7. Max Pooling

In the shallow depth (190th layer), the input image size is 60 × 60 with 512 filters (Figure 10g(i)). In the middle depth (252th layer), the input image size is 8 × 8 with 512 filters (Figure 10g(iii)). In the deep layer (354th), the image size is 8 × 8 with 512 filters (Figure 10g(v)). The learnable feature which is the response of the maximum (max) pooling layer to an input termed “activation”, as shown in Figure 10g(ii,iv,vi).

#### 3.3.8. Addition Layer

In the shallow depth (55th layer), the input image size is 120 × 120 with 64 filters (Figure 10h(i)). In the middle depth layer (138th), the input image size is 60 × 60 with 128 filters (Figure 10h(iii)). In the deep depth layer (336th), the image size is 8 × 8 with 512 filters (Figure 10h(v)). In the final addition layer (402th), the image size is 1 × 1 with 256 filters (Figure 10h(vii)). The learnable feature is the response of the addition layer to an input, as shown in Figure 10h(ii,iv,vi,viii).

#### 3.3.9. Depth Concatenation

In the shallow depth layer (56th), the input image size is 120 × 120 with 192 filters (Figure 10i(i)). In the middle depth (189th layer), the input image size is 60 × 60 with 512 filters (Figure 10i(iii)). In the deep depth layer (203th), the image size is 15 × 15 with 512 filters (Figure 10i(v)). In the final addition layer (307th), the image size is 15 × 15 with 512 filters (Figure 10i(vii)). The learnable feature, that is, the response of the depth concatenation layer to an input termed “activation”, as shown in Figure 10i(ii,iv,vi,viii).

#### 3.3.10. Concatenation and Dropout

In the deep layer, the image size of the concatenation layer (203th) is 1 × 2 with 256 filters (Figure 10j(i)). In the dropout layer (404th), the image size is 1 × 2 with 256 filters (Figure 10k(i)). This layer also removes 10% of the unused neurons from the network. The learnable feature is the response of the concatenation and dropout layers to an input, as shown in Figure 10j(ii) and 10k(ii), respectively.

#### 3.3.11. Group Normalization

The image size of the group normalization layer (278th layer) is 1 × 1 with 1024 filters (Figure 10l(i)). At layer 382th, the image size is 1 × 1 with 1024 filters (Figure 10l(iii)). The learnable feature is the response of the group normalization layer to the input, as shown in Figure 10l(ii,iv).

#### 3.3.12. Global Average Pooling

We use global average pooling before the fully connected layer to reduce the network dimension. At the 414th layer, the image size is 1 × 1 with 64 filters (Figure 10m(i)). The learnable feature, that is, the response of the global average pooling layer to the input, is shown in Figure 10m(ii).

#### 3.3.13. Fully Connected Layer

The fully connected layer (415th layer) connects all the neurons in the previous layer, then combines all the features learned by the previous layer to identify larger patterns to classify the images. The image size of this layer is 1 × 1 (Figure 10n(i)) with five stress conditions. The learnable features (indicated in black and gray) are the response of the fully connected layer to the input, as shown in Figure 10n(ii).

#### 3.3.14. Softmax Layer

The softmax function normalizes the input in the channel dimension and is used for the probability distributions of the output with a scale ranging from 0 to 1. The output of this layer is 1 × 1 (Figure 10o(i)) with 5 outputs of the stress condition and their probability distribution. The learnable feature (indicated in black) is the response of the softmax layer to an input, is shown in Figure 10o(ii).

## 4. Discussion

We conducted a live demonstration in the orchard of the National Institute of Fruit and Tea Science in Tsukuba, Japan (June, 2021), and deployed the HortNet417v1 network using a laptop equipped with the BUFFALO wide-angle web camera BSW200MBK (1920 × 1080 pixels: 30 fps) and Matlab R2020a. Twenty three plants were randomly selected, of which 19 were pot-cultivated and four were soil-grown in the open field. The network detects the water stress status of peach trees in real time with an accuracy rate of 73%. However, under the same conditions, MobileNetv2 shows an accuracy of 75%, which is slightly higher than the HortNet417v1 network. We generated a t-SNE plot (Figure 11, Figure 12 and Figure 13) to visualize the features of the first convolution, final convolution, and softmax activation layers and to evaluate the distance between various water-stress conditions of the pot-cultivated peach plants. The occlusion sensitivity in Figure 14 shows the change in the probability of belonging to the right stress condition, and also shows how the network learns features from the training data.

### 4.1. Network Stress Condition Prediction Accuracy Evaluation Using the TSNE Algorithm

Figure 11a visualizes the different intensities of the validation dataset distribution in the first convolution, final convolution, and softmax activation layers of the HortNet417v1 network. The majority of the stress conditions are separated, as shown in Figure 11b, while low stress data overlapped with moderate and high stress data. Figure 11c visualizes the different intensities of the test dataset distribution in the first convolution, final convolution, and softmax activation layers of the HortNet417v1 network. The majority of the stress conditions are separated, as shown in Figure 11d, while low stress data overlapped slightly with moderate and high stress data. This overlap is what we have observed in the validation and test data distribution and is caused by low, moderate, and high stress conditions. The images of these conditions are like to the images of the network and misclassified. By adding various patterns of the low, moderate, and high stress conditions, it is possible to solve this problem.

### 4.2. Exploration of Observations in the t-SNE (t-Distributed Stochastic Neighbor Embedding) Plot

The cross-validation classification in the last layer (output) is shown in Figure 12. The circle in the figure represents observation number 346, which provides the successful classification of the visualization of the cross-validation result of the 346th observation image from the validation data (actual and predicted stress conditions: high stress). This is difficult for the unexperienced human eye to detect. However, observer number 3 shows the misclassified visualization (actual: high stress; predicted: no stress) of the 3rd image from the validation dataset.

The classification ability of the output is shown in Figure 13. The circle in the figure shows observation number 30, which provides the successful classification (actual and predicted stress condition: high stress) of the visualization of the 30th image from the test data. This is also difficult for the unexperienced human eye to detect. However, observation number 378 shows the misclassified visualization of the 376th image from the test dataset (actual: high stress, predicted: no stress). The misclassification occurred due to the network’s inability to learn the pattern differences among low, moderate, and high stress conditions.

### 4.3. Predicted Result Evaluation Based on Occlusion Sensitivity and the LIME (Locally Interpretable Model-Agnostic Explanation) Technique

The occlusion sensitivity of all images and the red color in LIME indicate the most important areas of the image that the network uses for classification decisions (Figure 14a (ii, iii; top and bottom). Figure 14a shows that the network predicted a high-stress condition with a probability of 0.90 and 0.59, respectively. The occlusion sensitivity (Figure 14a(ii)) indicating the reason for the prediction is in the middle part for IMG_0621HS and the right corner of the lower part for IMG_5267HS. However, the LIME (Figure 14a(iii)) indicates that the reason for the prediction is in the left corner of the top part for IMG_0621HS and the middle to left corner of the lower part for IMG_5267HS.

In Figure 14b(i; top, bottom), the network predicted the low stress condition with a probability of 0.98 and 0.91, respectively. The occlusion sensitivity (Figure 14b(ii)) indicating the reason for the prediction is shown in the middle of the top part (background) for IMG_4357LS. However, occlusion sensitivity indicates that the reason for the prediction is in the left and right corners of the top part and the right corner of the lower part for IMG_0993LS. The LIME (Figure 14b(iii)) suggests that the reason for the prediction is in the middle (mostly background) of the top part for IMG_0621HS and the left corner of the top part for MG_5267HS. Both the occlusion sensitivity and the LIME technique shows that the features of IMG_4357LS are learned incorrectly. This can be solved by adding more LS training data.

In Figure 14c(i; top, bottom), the predicted probability of the moderate stress condition is 0.91 and 0.99, respectively. The occlusion sensitivity (Figure 14c(ii)) indicates the reason for the prediction is in the right corner of the lower and top part, respectively, for IMG_2983MS. For image IMG_5029MS, the network also shows that the reason for the prediction is in the bottom of the lower part. In both cases, the right corner of the top part (background) for IMG_2983MS and the bottom of the lower part for IMG_5029MS indicate that the features are learned incorrectly. The LIME (Figure 14c(iii)) suggests that the reason for prediction is in the left, middle, and right parts for IMG_2983MS and the middle part for IMG_5029MS. In this case, the LIME technique indicates a more accurate position than the occlusion sensitivity.

In Figure 14d(i; top, bottom), the network predicts the no stress condition with a probability of 0.86 and 0.98, respectively. The occlusion sensitivity (Figure 14d(ii)) indicates the reason for the prediction is the right corner of the lower part for IMG_2721NS and the middle of the top part for IMG_2586NS. LIME (Figure 14d(iii)) shows that the reason for the prediction is mainly in the middle and slightly right parts of IMG_2721NS. However, for image IMG_2856NS, the network shows the left and right corners of the top and middle of the lower part. In both IMG_2721NS and IMG_2856NS, the LIME technique also shows a slightly inaccurate position (lower and top right) as compared to the occlusion sensitivity.

In Figure 14e(i; top, bottom), the prediction probability for the very high stress condition is 0.71 and 0.98, respectively. The occlusion sensitivity (Figure 14e(ii)) indicates the reason for the prediction is in the right corner of the top part for IMG_0241VHS and the middle to the left of the lower part for IMG_1399VHS. LIME (Figure 14e(iii)) shows that the main reason for the prediction is in the middle of the lower half of IMG_0241VHS and the left corner of the lower half of IMG_1399VHS.

## 5. Conclusions

This article mainly describes the HortNet417v1 architecture and provides network performance results by evaluating various performance indicators. We use the most descriptive approach to prove that HortNet417v1 can classify various water-stress conditions of the pot-cultivated peach plants. The main findings are as follows:
− Classification of uneven data sets under various stress conditions, which may lead to lack of information and diversity of images and stress conditions. Most pre-trained networks converge with higher accuracy after 25 epochs but HortNet417v1 requires 36 epochs and more time to achieve higher accuracy. This response is because the weight of the pretrained model (Xception, ShuffleNet, and MobileNetv2) which is trained with millions of images, when actuated on a new training dataset, can converge at a faster rate than a network like HortNet417v1 in which network weights are randomly initialized instead of inherited from the previous model.

The research directions we are about to proceed along are as follows:
− In this experiment, we collect image data through a handheld mobile phone. In our next experiment, we will use some other fixed imaging platform surrounding the target plant to capture more time series data under various stress conditions and thus will improve the image data diversity and imbalance of the data amount between the stress conditions.− Since the development of the network is a continuous process, the authors plan to modify the network structure, optimize the network hyperparameters, and train the network with more data to improve the prediction accuracy in real time. Then, this technology makes it possible to extend the study to a large agricultural area, not only for peach trees, but also for other types of fruit tree.

## Figures and Tables

**Figure 1 sensors-21-07924-f001:**
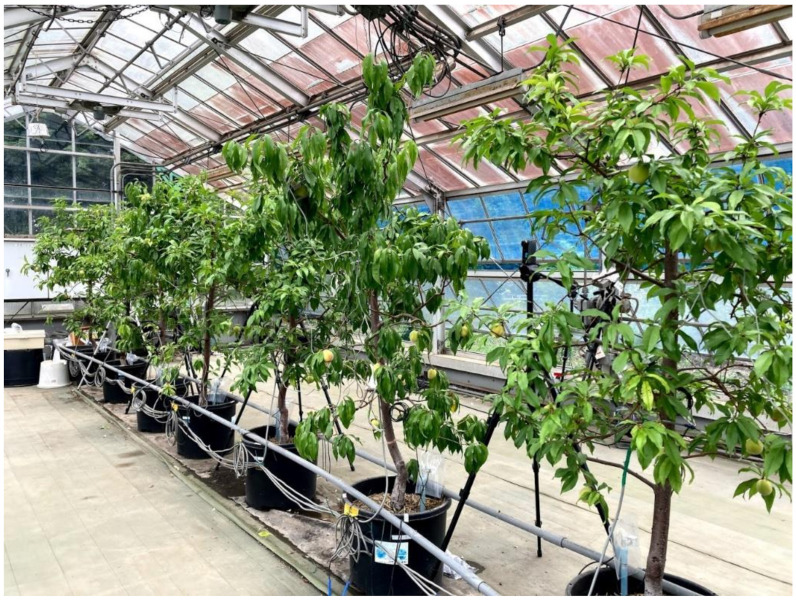
The planted peach tree cultivation system.

**Figure 2 sensors-21-07924-f002:**
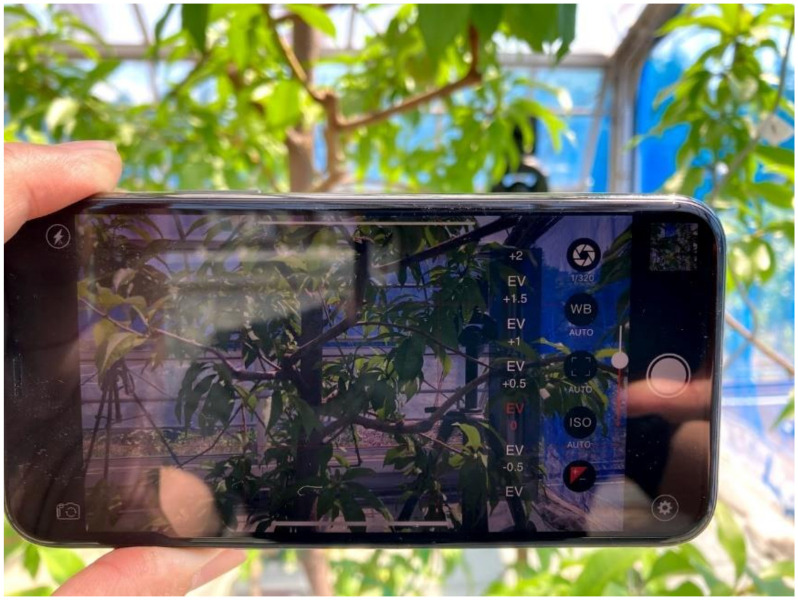
Image-acquisition process.

**Figure 3 sensors-21-07924-f003:**
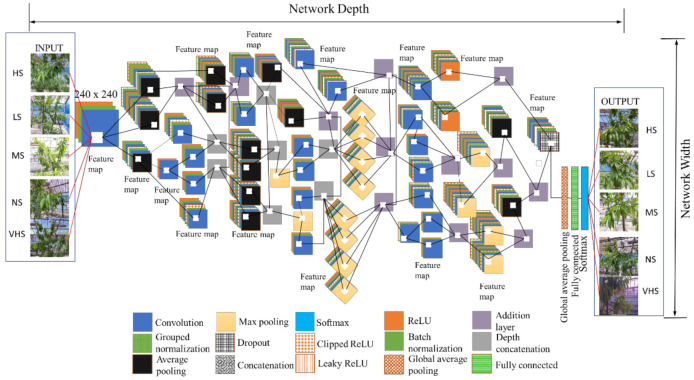
HortNet417v1 network architecture (HS—high stress, LS—low stress, MS—moderate stress, NS—no stress, VHS—very high stress).

**Figure 4 sensors-21-07924-f004:**
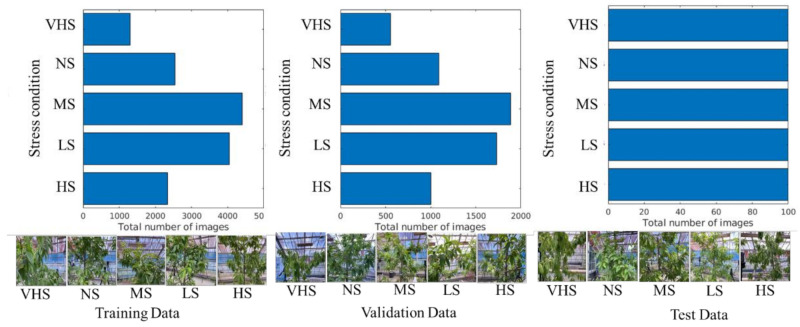
Visualization of the input image dataset.

**Figure 5 sensors-21-07924-f005:**
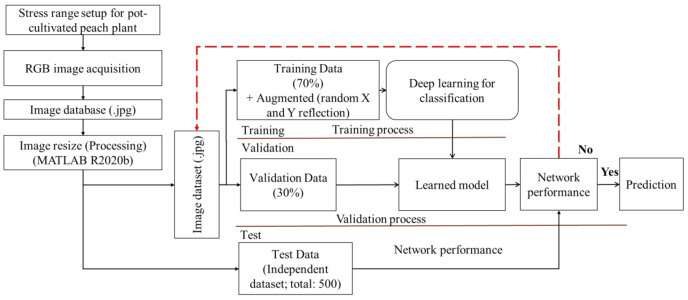
Schematic diagram of the network image dataset preparation and analysis.

**Figure 6 sensors-21-07924-f006:**
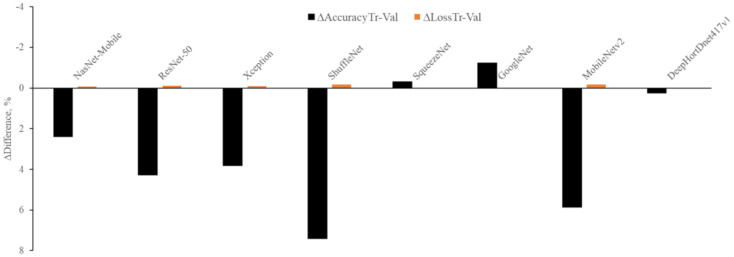
Visualization of network stability.

**Figure 7 sensors-21-07924-f007:**
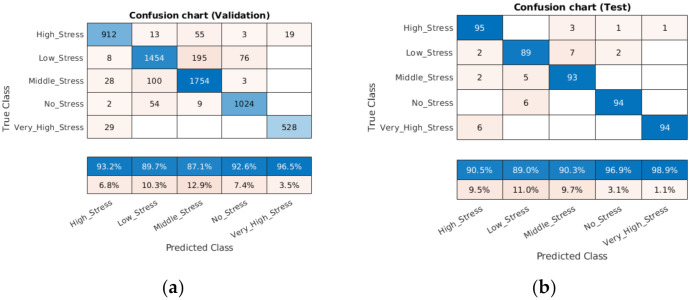
Confusion matrix with validation (**a**) and test (**b**) datasets.

**Figure 8 sensors-21-07924-f008:**
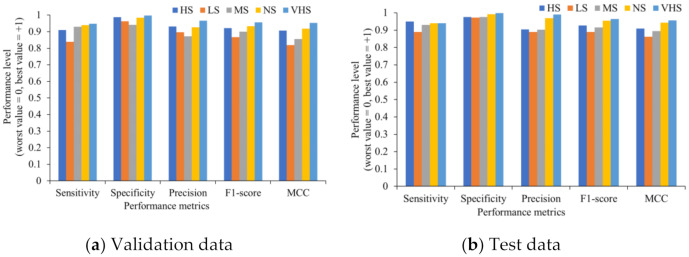
Evaluation of the individual stress condition.

**Figure 9 sensors-21-07924-f009:**
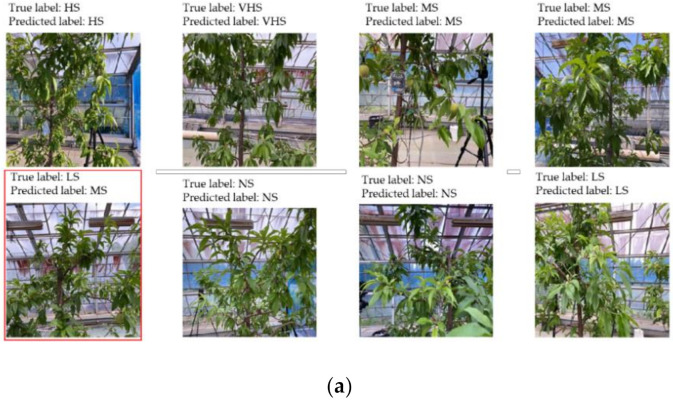
Prediction based on validation (**a**) and test (**b**) image datasets.

**Figure 10 sensors-21-07924-f010:**
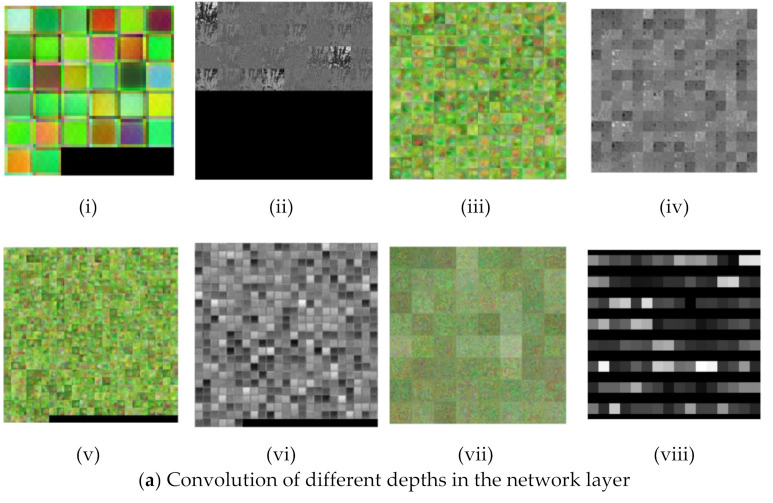
Visualization of network features and layer activations at different depths of the network.

**Figure 11 sensors-21-07924-f011:**
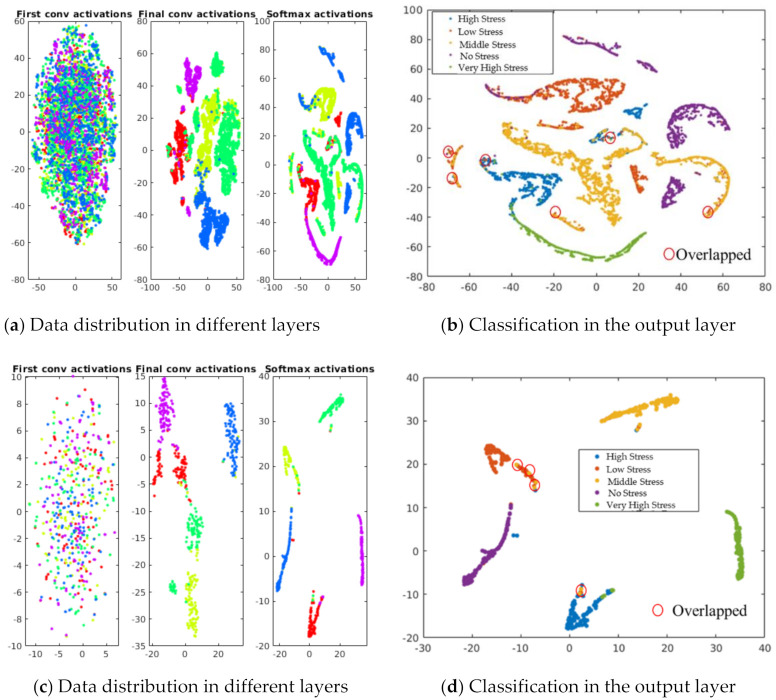
Visualization of data distribution using t-SNE (t-distributed stochastic neighbor embedding) algorithm.

**Figure 12 sensors-21-07924-f012:**
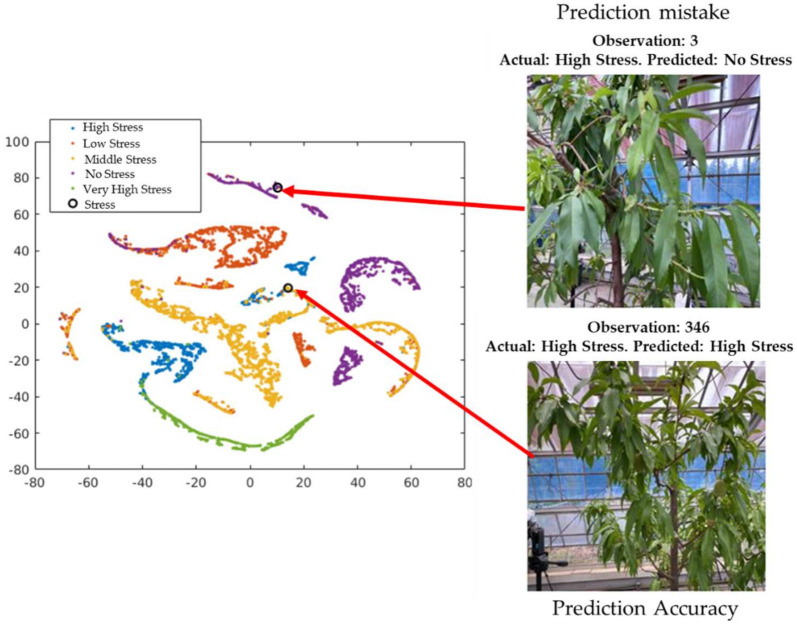
Exploration of observations in the t-SNE plot based on validation data.

**Figure 13 sensors-21-07924-f013:**
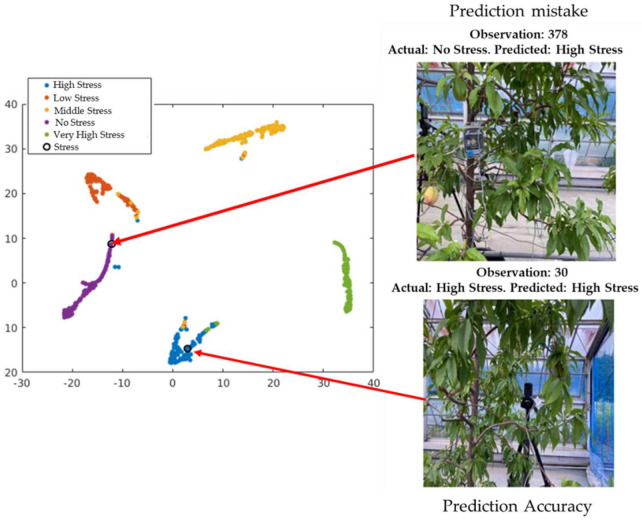
Exploration of observations in the t-SNE plot based on test data.

**Figure 14 sensors-21-07924-f014:**
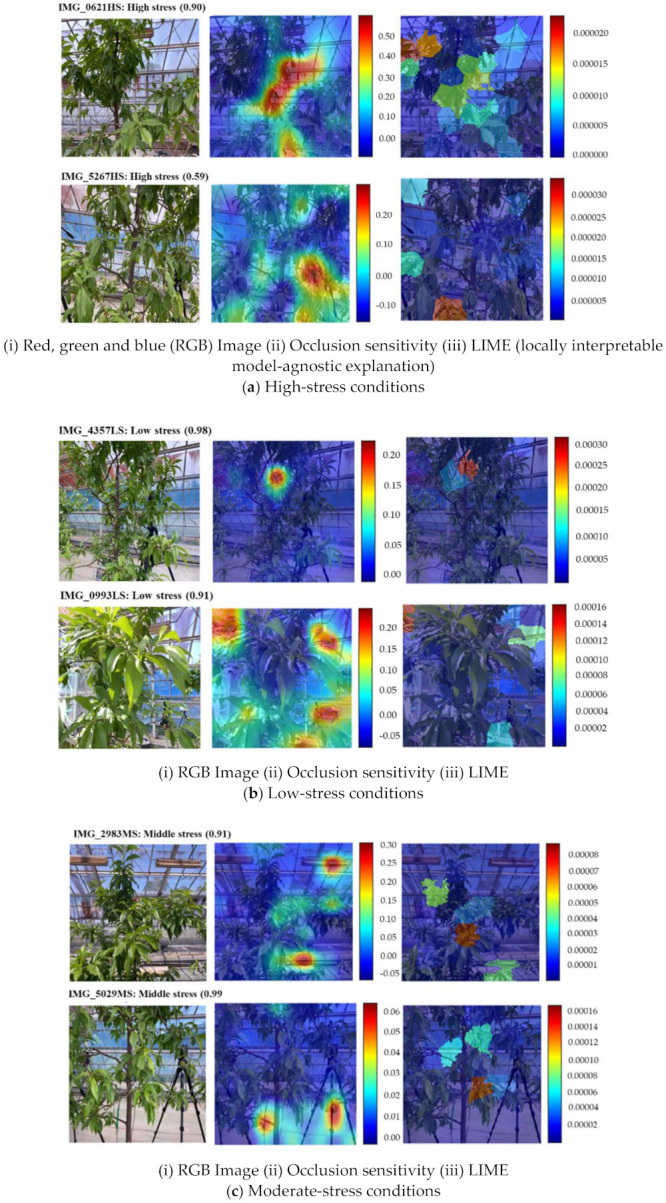
Visualization of the HortNet417v1 network decision behind the prediction of classification of the peach plant stress condition.

**Table 1 sensors-21-07924-t001:** Statistics of the HortNet417v1 architecture.

Layer Name	Total Number of Layers	Layer Name	Total Number of Layers
Image Input	1	Group Normalization	6
Convolution	124	Addition layer	12
ReLU	65	Depth Concatenation	7
Clipped ReLU	31	Global Average Pooling	1
Leaky ReLU	25	Concatenation	1
Dropout (10%)	1	Fully Connected	1
Batch Normalization	112	Softmax	1
Average Pooling	14	Pixel classification (Output)	1
Max Pooling	14		

**Table 2 sensors-21-07924-t002:** Network for comparative analysis.

Network	Number of Layers	Input Image Size	References
NasNet-Mobile	913	224 × 224 × 3	[26]
ResNet-50	177	224 × 224 × 3	[27]
Xception	170	299 × 299 × 3	[28]
ShuffleNet	172	224 × 224 × 3	[29]
SqueezeNet	68	227 × 227 × 3	[30]
GoogleNet	144	224 × 224 × 3	[31]
MobileNetv2	154	224 × 224 × 3	[32]
HortNet417v1	417	240 × 240 × 3	-

**Table 3 sensors-21-07924-t003:** Training performance (comparative analysis).

Network	Time (Min)	Max Epoch	TA (%)	VA (%)	TeA (%)	TL (%)	VL (%)
NasNet-Mobile	225.58	25	98.50	96.10	96.80	3.00	11.00
ResNet-50	26.35	25	98.85	94.56	94.00	4.00	16.00
Xception	29.23	24	100	96.18	97.20	2.00	11.00
ShuffleNet	23.18	22	100	92.60	93.60	3.00	21.00
SqueezeNet	6.54	27	58.85	59.16	62.00	89.00	87.00
GoogleNet	1.31	3	28.85	30.08	20.00	15.50	15.20
MobileNetv2	19.3	26	100	94.13	95.40	2.00	20.00
HortNet417v1	213	36	90.77	90.52	93.00	21.00	20.00

## Data Availability

Data will be available upon completion of project and publication of the project report upon request to the corresponding author. Any request will be reviewed and approved by the sponsor, NARO, intellectual property department, researcher, and staff on the basis of the absence of competing interest. Once approved, data can be transferred after signing of a data access agreement and confidentiality agreement.

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
