# Peer review of "HortNet417v1—A Deep-Learning Architecture for the Automatic Detection of Pot-Cultivated Peach Plant Water Stress"

_sensors, 2021, doi:10.3390/s21237924_

Round 1

Reviewer 1 Report

The author presented a study to utilize deep learning to measure the plant water stress for greenhouse-grown peach trees. A deep learning model HortNet417v1 was trained and tested on 50,562 images for recognition, classification, and visualization of plant stress conditions, such as no stress, low stress, moderate stress, high stress, and very high stress. The developed method reported test accuracy as 90.77% training, 90.52% cross-validation, and 15 93.00% without any overfitting issue. irrigation water management is essential for plant growth and productivity. This paper presents a stimulating research topic and is of interest to the wide scientific community and growers as this approach could assist in irrigation decision-making. The authors explained the methodology in detail and discussed the results comprehensively. In general, the manuscript is well written, however, I have a few comments for the author’s consideration.

Though the authors presented the results comprehensively, the discussion part in the paper is weak. The authors should add more discussion in the paper. Getting the results is important, but the interpretation of the results is more important on how these are obtained and could be used for further studies. I suggest including more discussion with the results section. Also, add the limitation of the proposed method as well as some future research directions.

I suggest revising the conclusion section. The methodology is already discussed, no need to present it in the conclusion again. Similarly, the results are not required to be repeated in the conclusions. The authors should focus on presenting the key findings/conclusions of the work with proposed future research directions in a small paragraph.

Another weakness of the paper is the repetition of the wording/sentences throughout the manuscript. For example, the results for validation and test are different but presented in similar-looking sentences and it can be seen throughout the paper and similar for the other results (layers). The author should revise the writing style to remove the repetition of sentences, even if reduces the paper length. Repetition of sentences is taken negatively by readers.

Line 75: Figure 1 is not HortNet417v1 architecture

Line 262: Define the terms used in the table

Line 292: Revise the figure, change the axis title and label font color to black

Figure 14: Caption is missing, check for other figures too. Caption should be placed right after the figure.

One suggestion would be to remove “the author (M. P. I.)” from the manuscript text (Introduction and Conclusion). The reader would know who the author is from the paper’s first page.

For all the abbreviations, use the full name for the first time the term is used in the paper.

The authors used both past and present tenses throughout the paper. English editing is recommended for grammar correction.

Reviewer 2 Report

The new architecture proposed in this paper does have practical innovations as a whole. However, the English expression of this article is too many long sentences, which is not concise and clear enough to read. Secondly, the chart format of this paper is not fine enough and needs to be adjusted.

In figure 2. In the process of acquisition, is it possible to use a fixed mobile platform to realize image acquisition, so as to reduce the image error of manual acquisition?

In figure3. 
Fig.3 Layout of the sample module is suggested to be adjusted again to be as intuitive and concise as possible. Should the text annotations in the same picture be consistent?

In table 1.
The data in the table can be evenly distributed vertically to make it look more professional, right?

Line 191-209. 
You may re-think of a theoretical formula that's too general and you don't have to go into detail, right?

In figure 8.

The comparison diagram should be adjusted to the same standard size.

In figure9.

When multiple pictures are placed in a picture and multiple annotations are required, can the same annotation be annotated accordingly? For example, Low stress is marked as L to make the article more readable.

In figure 14.

Parameter standard bar should be consistent.

In the conclusion part, the advantages of the deep network architecture proposed in this paper can be further highlighted. The abstract part of this paper points out that this paper can effectively reduce the problem of overfitting. Should it be reflected in the summary part?

Round 2

Reviewer 1 Report

The authors have revised the manuscript as I pointed. Some minor suggestions for author's consideration. 

  1. The newly added paragraph (most part) in the discussion is the repetition of methodology. Could remove those particular sentences.
  2. All authors should carefully read the paper for necessary corrections. The paper still has many instances of long sentences and incorrect grammar. 

Author Response

1. The newly added paragraph (most part) in the discussion is the repetition of methodology. Could remove those particular sentences.

Answer: Modified

2. All authors should carefully read the paper for necessary corrections. The paper still has many instances of long sentences and incorrect grammar. 

Answer: Modified
